# Gametocyte-specific and all-blood-stage transmission-blocking chemotypes discovered from high throughput screening on *Plasmodium falciparum* gametocytes

Giacomo Paonessa[1,5], Giulia Siciliano[2,5], Rita Graziani[1], Cristiana Lalli[1], Ottavia Cecchetti[1], Cristina Alli[1], Roberto La Valle[2], Alessia Petrocchi[3], Alessio Sferrazza[3], Monica Bisbocci[1], Mario Falchi [4], Carlo Toniatti[1,3], Alberto Bresciani [1✉] & Pietro Alano [2✉]

Blocking *Plasmodium falciparum* human-to-mosquito transmission is essential for malaria elimination, nonetheless drugs killing the pathogenic asexual stages are generally inactive on the parasite transmissible stages, the gametocytes. Due to technical and biological limitations in high throughput screening of non-proliferative stages, the search for gametocyte-killing molecules so far tested one tenth the number of compounds screened on asexual stages. Here we overcome these limitations and rapidly screened around 120,000 compounds, using not purified, bioluminescent mature gametocytes. Orthogonal gametocyte assays, selectivity assays on human cells and asexual parasites, followed by compound clustering, brought to the identification of 84 hits, half of which are gametocyte selective and half with comparable activity against sexual and asexual parasites. We validated seven chemotypes, three of which are, to the best of our knowledge, novel. These molecules are able to inhibit male gametocyte exflagellation and block parasite transmission through the Anopheles mosquito vector in a standard membrane feeding assay. This work shows that interrogating a wide and diverse chemical space, with a streamlined gametocyte HTS and hit validation funnel, holds promise for the identification of dual stage and gametocyte-selective compounds to be developed into new generation of transmission blocking drugs for malaria elimination.

[1] Department of Translational and Discovery Research, IRBM S.p.A., Pomezia, Roma, Italy. [2] Dipartimento di Malattie Infettive, Istituto Superiore di Sanità, Roma, Italy. [3] Department of Drug Discovery, IRBM S.p.A., Pomezia, Roma, Italy. [4] Centro Nazionale AIDS, Istituto Superiore di Sanità, Roma, Italy. [5] These authors contributed equally: Giacomo Paonessa, Giulia Siciliano. ✉email: a.bresciani@irbm.com; pietro.alano@iss.it

In 2021, the World Health Organization (WHO) "World malaria report 2021"[1] reported 241 million new malaria cases and over 600.000 deaths, mainly due to *Plasmodium falciparum*, confirming that progresses in controlling malaria, reached between 2000 and 2015, have stalled in the past five years. This led the WHO to adopt a new strategy to accelerate progresses towards the long-term goal of malaria elimination. In fact, malaria elimination is an objective set in the WHO "Global Technical Strategy for Malaria 2016–2030" in the Strategic Framework Pillar to "Accelerate Efforts Toward Elimination" (WHO Global Technical Strategy for Malaria 2016–2030)[2].

In malaria, symptoms are caused by the pathogenic asexual stages of the Plasmodium protozoan parasite that infect and proliferate within red blood cells able to adhere to the microvasculature endothelium of virtually all internal organs. Instead, transmission of the parasite to *Anopheles* mosquito vectors relies on gametocytes: non-dividing, highly differentiated Plasmodium sexual stages that, in the case of *P. falciparum*, mature in red blood cells in 10–12 days through five morphological stages (I–V). While circulating, mature gametocytes are taken in the mosquito blood meal, in which they transform into male and female extracellular gametes. The mating of the gametes in the mosquito gut generates a fertilized zygote that eventually produces thousands of sporozoites. These are passed on from the insect salivary glands to humans at the mosquito bite.

With the aim of reaching malaria elimination, Medicines for Malaria Venture (MMV) proposed a set of revised target product profiles (TPP) to treat and prevent malaria[3]. Besides the search for new molecules to treat the acute phase of malaria, and to face the constant threat of the onset of parasite drug resistance (target candidate profile 1, TCP1)[3], the focus is put on the discovery of drugs able to prevent infections by blocking parasite transmission from infected individuals to the mosquito vectors (target candidate profile 5, TCP5)[3]. This goal can be achieved by compounds targeting the *Plasmodium* gametocytes.

Except for primaquine, which however has safety liabilities as it can cause haemolysis in individuals with glucose-6-phosphate dehydrogenase deficiencies[4], current antimalarial drugs fail to kill *P. falciparum* mature gametocytes[5]. This is likely due to the different biology underlying sexual *vs* asexual stage development, suggesting that future anti-gametocyte drugs may have to hit gametocyte specific mechanism(s) of action. Nevertheless, dual active drugs, ideally active with comparable potency on both stages, are currently considered highly desirable by several malaria drug discovery initiatives, e.g., MMV. Example of compounds being investigated are the spiroindolone Cipargamin/KAE609, targeting the PfATP4 cation ATPase, the 2-Aminopyridine MMV048, targeting the parasite phosphatidylinositol-4-kinase PI4K and the quinoline-4-carboxamide M5717/DDD498, active on the parasite translation/elongation factor 2[6]. Importantly, potential gametocyte specific drugs will not cause the selection of parasite resistant genotypes during asexual proliferation, as predictable for dual active drugs, thus securing a longer lifetime.

So far, the cumulative amount of compounds tested against *P. falciparum* gametocytes was less than 300,000 in several independent screening[7–17], compared to the over 5 million that were screened against the asexual stage parasites, half of which in only two screenings[18]. This is partly due to the challenge of reliably measuring the viability of non-proliferative gametocytes in a sensitive and robust high throughput screening (HTS) assay format. A variety of HTS gametocyte assays were used so far relying on multiple detection techniques: fluorescent/luminescent reporters[19,20], parasite enzymes activity assays[21], gametocyte ATP content determination[22], uptake of redox sensitive dyes[23] and the assessment of the motility of male gamete *flagella*[24]. In these approaches, one or more protocol steps are not amenable to large screenings or may affect gametocyte response to treatment. Some examples are: the need to purify gametocytes from uninfected erythrocytes[7,22], gametocyte treatment in the absence of human serum[8] and the use of technically demanding imaging procedures[13,19]. These have altogether prevented to establish screening approaches for anti-gametocyte molecules able to interrogate a chemical space of a size like the one explored on the asexual blood stages.

To this end, here we report the establishment of a simple, sensitive and robust HTS assay funnel for the identification of anti-gametocyte compounds including set of counter-screenings and validation assays. With this tool we set out to identify bona fide gametocyte active chemotypes and investigate the selectivity of them against the human host and the asexual stage parasites.

## Results

**Gametocyte assay optimization and miniaturization.** In order to screen the CNCCS (the Italian national chemical collection) compound library for molecules able to inhibit *P. falciparum* gametocyte viability, we optimized and miniaturized an assay using the genetically modified parasite line 3D7*elo1-pfs16*-CBG99[20,25]. In this line, the CBG99 luciferase-coding region (from the click beetle *Pyrophorus plagiophthalamus*) is integrated into the parasite genome and expressed under the control of the *pfs16* gametocyte-specific promoter. Gametocytes of this line produce a strong and gametocyte-specific bioluminescent signal with optimal stability. The highest reporter signal is reached in stage IV–V gametocytes, with optimal signal to background (S/B) ratios[25]. According to our previous work, the use in the luciferase assay of a non-lysing, ATP-free, D-luciferin substrate was adopted to more reliably measure gametocyte viability, as confirmed by single gametocyte bioluminescence imaging[20].

Obtaining an accurate gametocyte synchronization with minimal parasite handling is of paramount importance to establish a robust and reliable HTS assay. To this aim, the timing of addition of N-acetyl glucosamine (NAG), used to clear residual asexual stages after appearance of stage I gametocytes, was optimized. A three-day treatment with 50 mM NAG, followed by an additional five days of cultivation, yielded cultures with about 2% gametocytaemia of early-stage V gametocytes, as confirmed by microscopy examination of Giemsa-stained blood smears (Fig. 1a). Using these cultures, different haematocrit percentages (0.625, 1.25 and 2.5%) and two incubation times (48- and 72 h) were tested to identify the optimal luciferase readout. Ten μM methylene blue (MB) was used as reference compound (Fig. 1b). As a result, 0.625% haematocrit at 48 h was found to be an acceptable compromise between an excellent signal to background (S/B) ratio (i.e., the ratio between the average of the vehicle treated *vs* MB treated cultures, a ratio greater than three is commonly considered appropriate, in this case it was 13.6) and a reasonable culture amount per well. Finally, using the optimized assay conditions, we established the final assay metrics and robustness by incubating half of a 384-well plate with 10 μM MB and half 384-well plate with vehicle dimethyl sulfoxide (DMSO) for 48 h, followed by luminescence detection. Results reported in Fig. 1c show that, in these conditions, we obtained a S/B of 53 and a Z' of 0.76[26]. In addition, the coefficient of variation% (CV%) of both positive and negative controls was less than 10% (Fig. 1c), that is commonly considered the threshold for optimal assay precision. The selected screening conditions not only combine both optimal biological conditions and good readout but are also straightforward because they do not require gametocyte purification steps. Finally, parasite culture volumes necessary to perform an entire screening can be easily managed: a total volume of 100 ml routine gametocyte culture (4% haematocrit, 1–2%

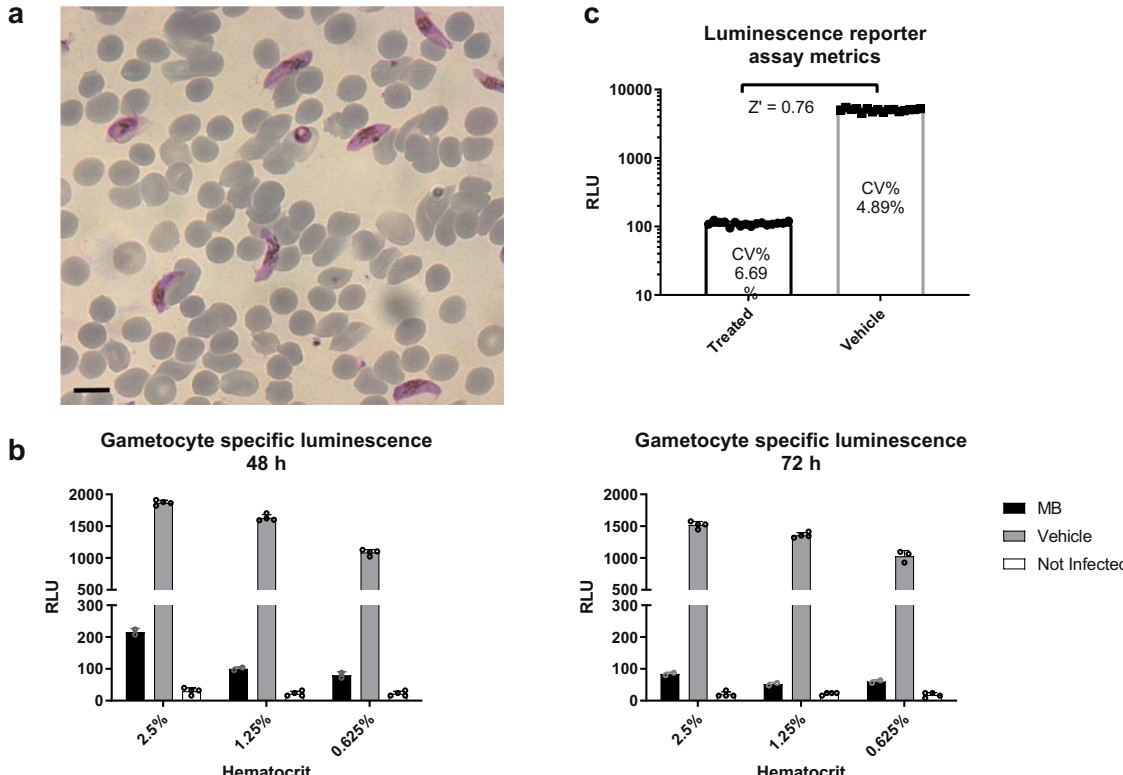

**Fig. 1 HTS assay optimization. a** Representative image of the gametocyte stages used in the HTS optimization: Giemsa-stained blood smears of infected red blood cell (RBC) cultivation after three-day treatment with 50 mM NAG followed by additional 5 days of cultivation. Early-stage V gametocytes are visible and a 2% gametocytaemia can be calculated. Black scale bar = 10 μm **b** Luminescence counts, expressed as Relative Light Units (RLU, averages and standard deviations are depicted, $n = 4$ independent experiments or $n = 2$ independent experiments for MB), of haematocrit dilutions of uninfected and of vehicle or 10 μM Methylene Blue (MB) treated infected RBC at 48- and 72 h incubation. Cultures were treated with NAG as above described. **c** Luminescence reporter assay metrics of MB treated vs. untreated culture at 48 h (averages and standard deviations are depicted, $n = 48$ technical replicates). Coefficient of variation percentage (CV%) are reported within each bar. Assay with a precision of CV% < 10 are considered optimal. For both **a** and **b** due to the high difference in magnitude between high and low conditions, a statistical analysis to assess significance was not needed.

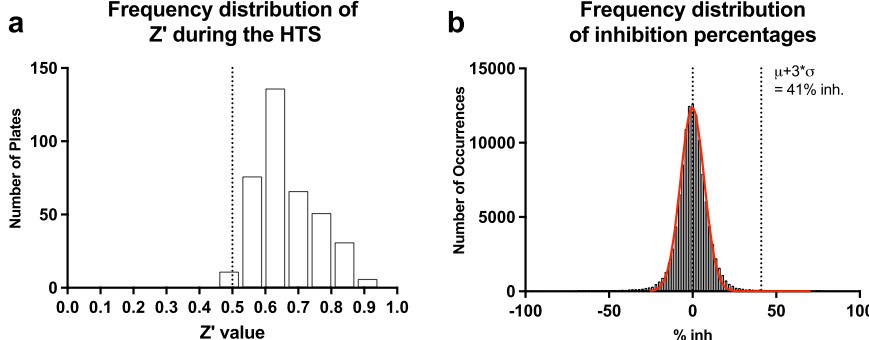

**Fig. 2 HTS Z′ value and inhibition frequency. a** Diagram of number of 384 wells plates grouped by the same Z′ value. In total, the Z′ value was calculated on 378 plates. **b** Inhibition percentage frequency distribution of the 119,059 compounds tested. The dotted line represents the 41% inhibition cut off limit.

gametocytaemia) was enough for an average screening batch of 19.200 data points. Altogether these results paved the way for the HTS campaign described here.

**High throughput screening and hit identification.** A collection of 119.059 compounds of the CNCCS library was tested on gametocytes of the 3D7*elo1-pfs16*-CBG99 strain at 10 μM using the protocol described above. The Z′ values were found to be greater than or equal to 0.5 for all screening plates indicating that the assay was sufficiently robust to test the compounds (Fig. 2a). The distribution of the compound activities converged to normal

(or Gaussian) distribution (Fig. 2b); therefore, compounds with an activity equal to or greater than the average activity plus three standard deviations (41% inhibition) were considered hit compounds. Applying these parameters, 960 compounds, corresponding to 0.81% of the total, were identified as active in the primary screening and subjected to confirmation assays.

**Hit confirmation.** To validate the active compounds the 960 hits were tested at three concentrations, namely 20, 4, and 0.8 μM, on the same 3D7*elo1-pfs16*-CBG99 strain used for the screening. Furthermore, they were also tested at 10 μM in a mammalian cell

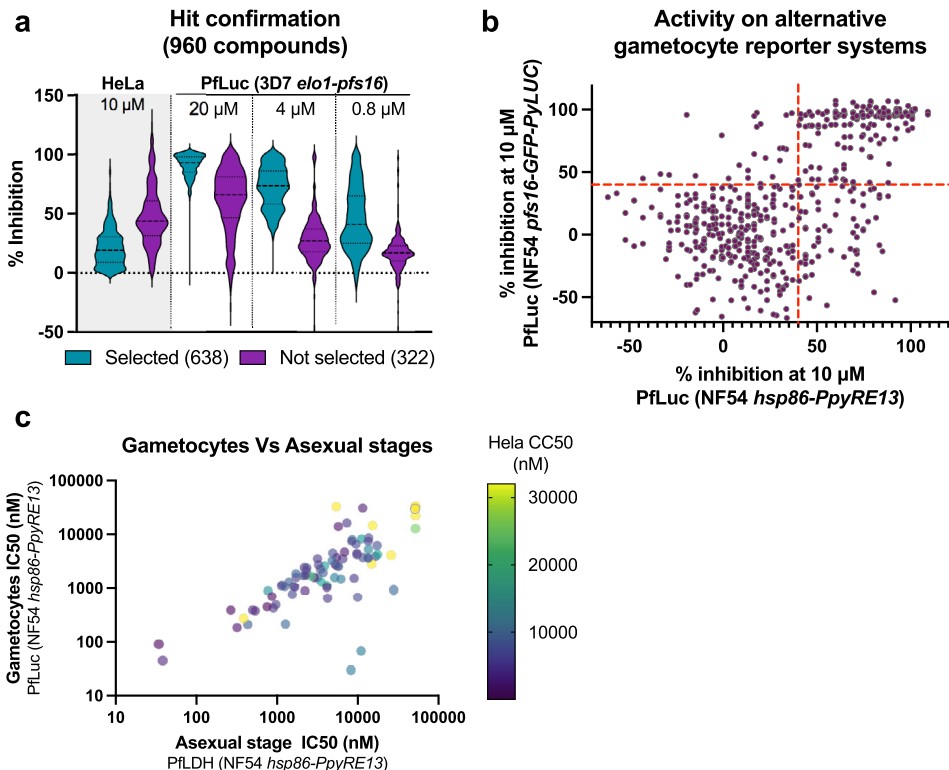

**Fig. 3 Hit confirmation and selectivity analysis.** The number of replicates for each experimental point was one for all the assays due to the technical challenges related to gametocyte assays. **a** Distribution chart representing the percentage of inhibition of 960 compounds in HeLa cells proliferation assay at 10 µM and in 3D7*elo1-pfs16*-CBG99 gametocyte assay at 20, 4, and 0.8 µM. The light blue and violet areas represent the selected and the discharged compounds, respectively. Dot horizontal line inside the areas indicates the median inhibition percentage (thick line) and top and bottom quartiles (thin lines). **b** Scatter plot of the percentage inhibition of PfLuc assays of 638 compounds on NF54 *pfs16*-GFP-PyLUC strain *vs* NF54 *hsp86*-PpyRE13 strain. Cross red lines, settled at 40% inhibition for each axis, divides the plot in four fields. **c** Bubble scatter plot (log–log) of the selected 84 compounds potency (nM) on asexual stages (pLDH assay) *vs* gametocytes (PfLuc assay) in the NF54 *hsp86*-PpyRE13 strain. The bubble colours correspond to the degree of potency of the HeLa proliferation assay coded by the column on the right.

line (HeLa) proliferation assay to exclude compounds with overt cytotoxicity.

Figure 3a shows a summary plot of these results: the cytotoxicity data at 10 µM and the gametocyte inhibition at 4 µM ruled the selection of 638 compounds for further profiling. To exclude false positive compounds that may interfere with the click beetle CBG99 luciferase signal and to confirm the activity on gametocytes produced by other *P. falciparum* lines, we tested the above 638 compounds at a single concentration of 10 µM on gametocytes from two NF54 strain derived lines: *pfs16*-GFP-PyLUC (*Photinus pyralis* luciferase under the same *pfs16* promoter)[14,16,27] and *hsp86*-PpyRE13 that was generated for this work to express the mutated *Photinus pyralis* PpyRE13 luciferase in gametocytes under the promoter of the *pfhsp86* gene.

As presented in Fig. 3b, 185 compounds (29%) were confirmed active in both lines (top right field), whereas 331 (52%) resulted poorly active in both lines (bottom left field). Assays with gametocytes of the NF54 *hsp86*-PpyRE13 line generally yielded more confirmed compounds (right top and bottom fields) than those with the NF54 *pfs16*-GFP-PyLUC line (top left and right fields). This strongly suggests that the inhibition of the luciferase reporter can represent a major confounding factor in the screening assay[28–31]. In fact, the NF54 *pfs16*-GFP-PyLUC line shares same *pfs16* promoter with the screening strain but differs in the reporter luciferase.

Altogether, this work suggested that differences between the line used in the HTS and those used in the confirmation assays (e.g., the different luciferase promoters, the different luciferase

enzymatic properties, such as pH dependence and thermal stability and other unrelated factors) may contribute to differential compound activities and confirmed the value of early adoption of confirmation assays to make sure to select valuable compounds only.

In order to avoid advancing too many similar compounds, we performed a clustering based on the Taylor Butina algorithm[32], a non-hierarchical clustering method that ensures that each cluster contains molecules with a certain cut-off (or threshold) distance from a central compound. Circular fingerprints with radius 2 and 2048 bits were generated using the RDKit software[33] with the purpose of generating a similarity matrix based on a Tanimoto index[34]. The effective number of neighbours for each molecule was calculated based on the Tanimoto level (0.8) used for clustering. This procedure gave a collection of 371 clusters, 205 of which were singletons. Subsequently, the selected set was subjected to quality control by LC-MS to check compound identity and purity (acceptable purity criteria set to be >90% peak area in the diode array trace). Altogether, the activity data on the three parasite lines, the compound clustering, the compound QC and medicinal chemist evaluation led to the identification of 84 molecules (Supplementary Data 2) that were progressed through the validation funnel. Among these, besides compounds that were active in all three gametocyte assay systems, we also included 15 compounds with interesting chemical properties that were active in the NF54 *hsp86*-PpyRE13 and 3D7*elo1-pfs16*-CBG99 strains only as we speculated that, being both the promoter and the reporter different between the two lines, they were likely true positives.

**Selectivity profiling**. With the selected 84 compounds, the subsequent step aimed at determining the gametocyte inhibition potency and the parasite stage selectivity between the gametocytes and the asexual stages in the NF54 $hsp86$-PpyRE13. $IC_{50}$ determinations showed that most compounds (64) were active against parasites in the micromolar range ($\geq 1\,\mu M$) while 21 were sub-nanomolar compounds. In terms of selectivity, 53 compounds (62%) had comparable potencies on gametocytes and on asexual stages, whereas 32 compounds (38%) had gametocyte selectivity values ranging from 2- to 268-fold (Fig. 3c). Similar data were obtained considering the gametocyte $IC_{50}$ values measured in the NF54 $pfs16$-GFP-PyLUC strain (see Supplementary Data 2).

In addition, we tested the 84 compounds in a dose-response fashion on HeLa cells to quantitatively monitor their cytotoxicity potential. This assay revealed that most compounds (51, 60%) were more active (greater than two-fold) on gametocytes than in HeLa, irrespectively to the anti-asexual stage or the anti-gametocyte activity (colour scale in Fig. 3c).

To further prioritize the 84 compounds, those with and $IC_{50}$ below $3\,\mu M$ in the NF54 $pfs16$-GFP-PyLUC gametocyte assay were selected. This set was further grouped in 14 groups and 29 singletons based on their chemical structure. Among these compounds, the elimination of those with a gametocyte/host selectivity less than five-fold and those with poor lead-like properties/tractability led to a final set of 30 molecules that were progressed to the male gamete exflagellation assay.

**Activity of selected compounds in mosquito parasite stage assays**. The above selected 30 compounds were subjected to a phenotypic male gamete exflagellation assay. This assay measures the viability of male gametocytes by quantitating the final step of maturation into motile microgametes in vitro[24]. This process naturally occurs in the mosquito gut within minutes from the blood meal. The acquisition of time lapse videos of gamete motility in bright field microscopy and the subsequent analysis to count exflagellation centres were performed in an automated HTS format in 384-well plates[13]. The compounds were tested in two independent biological replicates at the concentration of $1\,\mu M$. Methylene blue and compound DDD01035881 (compound 84)[13] were used as positive controls. The results of this study led to the identification of 16 compounds with anti-exflagellation activity (Fig. 4). Among those showing an inhibition of the male gamete exflagellation greater than 50%, nine were further progressed for testing their transmission blocking activity (Fig. 5a). The standard membrane feeding assay (SMFA) is used to measure the parasite transmission through mosquito. In this assay, mature stage V gametocytes of the *P. falciparum* strain NF54-$hsp70$-luc were treated with 1 and $10\,\mu M$ of the nine selected compounds for 24 h, added with human red blood cells and serum to achieve a haematocrit of 50% and then fed to mosquitoes. Luminescence was analyzed in the mosquito salivary glands 8 days post-feeding. Compound MMV0048 was used as positive control[35] and, as expected, it abolished the parasite transmission at both concentrations. With the exception of compound **64**, all compounds showed a high level of transmission inhibition at $10\,\mu M$. At the concentration of $1\,\mu M$, compound **53** was the most active, with about 100% inhibition, followed by compounds **13**, **16**, and **39** (Fig. 5b).

In summary, the screening of approximately 120,000 compounds against 3D7$elo1$-$pfs16$-CBG99 gametocytes led to the identification of 960 compounds with a hit rate of about 0.8%. Counter-screenings against HeLa cells and two alternative gametocyte systems (NF54 $hsp86$-PpyRE13 and NF54 $pfs16$-GFP-PyLUC), together with hit quality control and a chemical clustering analysis, restricted the set of hits to 84 molecules. The evaluation of

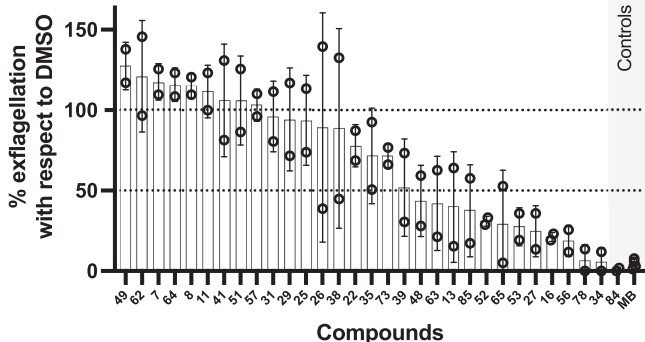

**Inhibibtion of male gametocyte exflagellation at 1 μM**

**Fig. 4 Male gamete exflagellation assay.** Open circles represent single experimental points. Averages and standard deviations are depicted for each condition ($n = 2$ independent experiments). Average DMSO exflagellation events per microscopy field: 17.6. Methylene Blue (MB) and compound DDD01035881 (84)[13] were used as positive controls.

**a** Compounds selected for SMFA

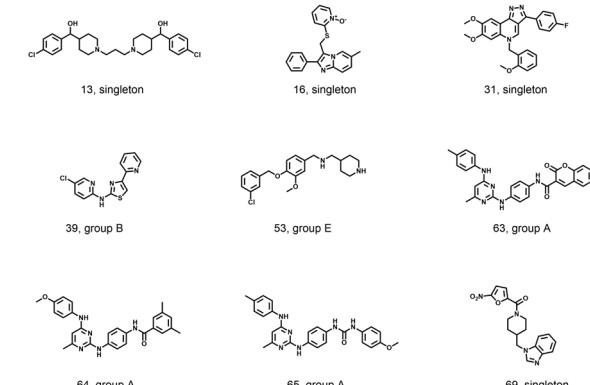

**b**

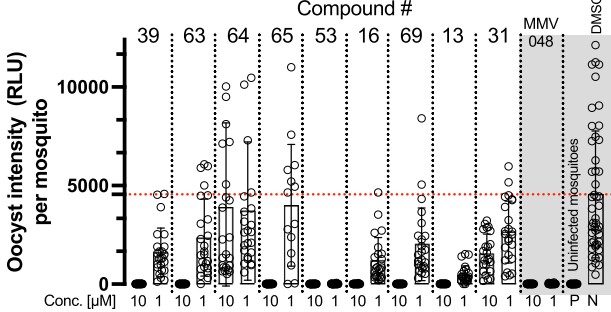

**Standard Membrane Feeder Assay (SMFA)**

**Fig. 5 *P. falciparum* transmission blocking activity to *Anopheles stephensi* mosquitoes of selected compounds by Standard Membrane Feeding Assays (SMFA). a** Structure of the 9 compounds selected for SMFA. **b** The graph shows the individual mosquito luminescence values used to determine oocyst intensity for DMSO (N), uninfected mosquitoes (P), positive control MMV0048 and the tested compounds at both concentrations. Averages and standard deviations are depicted for each conditions ($n > 14$ biological replicates).

the gametocyte *vs* asexual stage selectivity led to the selection of 30 compounds that were assayed in a male gamete exflagellation assay. Active molecules in this assay led to the final selection of nine compounds that were tested in the standard membrane-feeding

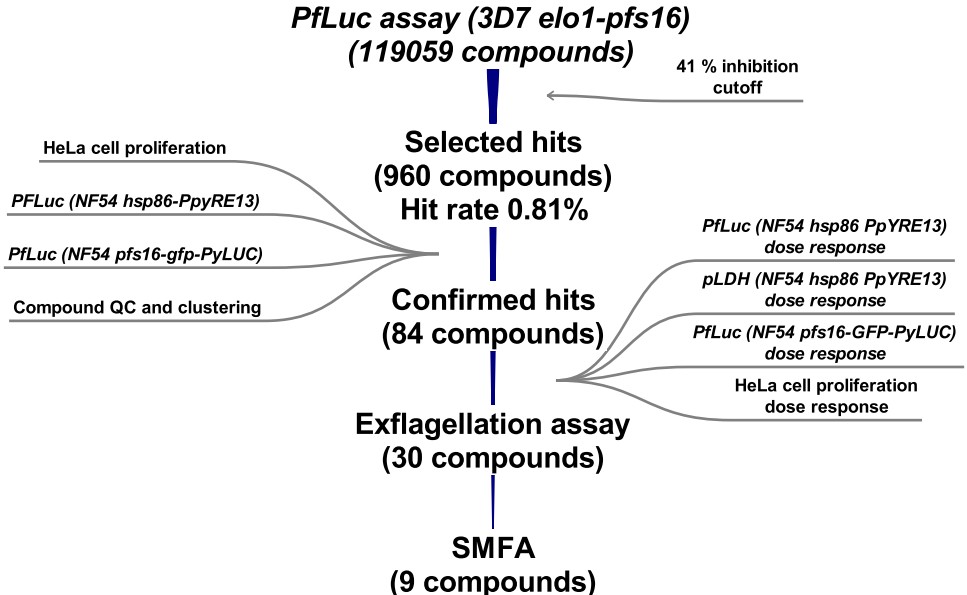

**Fig. 6 Screening funnel.** The CNCCS collection of approximately 120,000 was screened with gametocytes of the 3D7*elo1-pfs16*-CBG99 strain with a PfLuc assay at 10 µM concentration. To select the active compounds, a cut-off limit of 41% inhibition was applied resulting in 960 compounds (hit rate of about 0.8%). Three counter screening assays were performed on these compounds at fixed doses, a HeLa cell proliferation assay and two PfLuc assays on gametocytes of the NF54 *hsp86*-PpyRE13 and of the NF54 *pfs16*-GFP-PyLUC strains. An additional quality control assay and a chemical clustering analysis restricted the number of compounds to 84. These were filtered based upon potency as determined by full dose response assays complemented with a pLDH asexual stage assay on the NF54 *hsp86*-PpyRE13 strain. Results led to select 30 compounds to be tested in a male gamete exflagellation assay, whose results led to the final selection of 9 compounds tested in the standard membrane-feeding assay (SMFA), of which 8 turned out to be able to possess transmission blocking activity. Three of the latter represent, to our knowledge, novel chemotypes of *P. falciparum* transmission blocking molecules.

assay (SMFA). Finally, eight compounds turned out to have good *P. falciparum* transmission blocking activity and three of them are, to the best of our knowledge, novel chemotypes. These findings are discussed in the next section.

## Discussion

Ten years ago two milestone HTS campaigns populated the antimalarial drug discovery pipeline of hits against asexual blood stages[18], establishing a notable portion of the current antimalarial drug discovery portfolio. The size and diversity of the chemical space interrogated for asexual parasite killing activity undoubtedly represented a key factor in the success of that endeavour. These and other HTS campaigns contributed to the creation of the "Malaria Box", a set of 400 promising antimalarial leads and tools most of which, however, resulted to be poorly or at all active on mature gametocytes[19,36].

So far, compounds with anti-asexual stage activity are, in most cases, unable to kill other stages of parasite development, including gametocytes and liver stages. It has been speculated that the biology of the proliferating asexual parasites is somewhat different from that of the non-dividing, terminally differentiated gametocytes. This is thought to be particularly true for mature stage V gametocytes. On the other hand, the results of gametocyte HTS[13,36] led to the identification of both gametocyte-specific and dual active compounds, with the latter being quite frequent. This suggests that large anti-gametocyte HTS are likely to also discover novel anti-asexual stage compounds.

In this work, we developed a screening funnel (Fig. 6) for *P. falciparum* transmission blocking drug discovery starting with a simple, sensitive, robust and high-throughput assay followed by a set of counter-screening and validation assays to test the gametocyte specific activity of the identified molecules. We screened almost 120.000 compounds, more than one third of the number

globally interrogated so far in gametocyte HTS of diverse compound libraries[7–13,15–17].

The amplitude of the primary screening was made possible by the adoption of a luciferase-based *P. falciparum* gametocyte assay where the CBG99 luciferase reporter expression was driven by the sexual stage specific *pfs16* promoter. The screening protocol is completely homogeneous and uses non-purified gametocyte culture. To develop the signal, the simple addition of D-luciferine, with no added ATP, was required before luminescence measurement.

The validation of the 638 hits, confirmed on the primary assay, was performed using gametocyte assays relying on two structurally different luciferases driven by two different parasite promoters. The results revealed that a rather big portion of the hits was likely ascribed to a direct inhibition of the click beetle CBG99 luciferase enzyme. Differently from other screening funnels, relying on more complex and lower throughput assays, interrogating different gametocyte processes[14,37], the early and higher throughput use of these parasites lines conveniently and efficiently identified molecules to be directly progressed to more predictive phenotypic assays like male gamete exflagellation and SFMA.

In addition, this approach allowed for the early investigation of mature gametocyte *vs* asexual stage selectivity. We observed that 53 compounds (62%) had comparable potencies on gametocytes and on asexual stages, whereas 32 compounds (38%) had gametocyte selectivity values ranging from 2- to 268-fold. These results show that a HTS using gametocyte viability as primary readout can produce both dual active and gametocyte specific hits, suggesting that the parasite biology at least in part overlaps among different lifecycle stages.

In this work, to further prioritize and validate hit molecules, we used the male gametocyte viability assay that is universally adopted for its high predictivity of parasite infectiousness to mosquito. Male gametocytes are a minor fraction of the parasite

sexual stages, given the female biased sex ratio of *Plasmodium* gametocytes. In the NF54 genetic background used here, the typical male to female ratio is 1 to 10. Nevertheless, the efficient inhibition of male gametocytes is highly desirable for parasite transmission blocking drugs as these sexual stages appear to be critical in infections with very low gametocytaemias[38], a common feature in the asymptomatic carriers[39] who are ideal recipients of transmission blocking drugs. This choice may have led us to miss compounds active only on female gametocytes. However, we decided not to introduce a female gametocyte specific assay as the upstream luciferase gametocyte assays contained, according to the previously mentioned sexual bias, mainly female gametocytes. Hence, compounds that failed to inhibit exflagellation may be reconsidered in the future as possible female specific compounds.

The final SFMA validation allowed us to identify seven chemotypes able to block malaria parasite transmission through mosquitoes via their inhibitory activity on *P. falciparum* gametocyte viability. Four of these have been previously identified in other screening campaigns that used different cell-based assays on asexual parasites or gametocytes.

Compound 53 was previously identified in a HTS on asexual stages[40] and in a gametocyte HTS assay measuring the parasite ATP content[7] and a similar analogue was shown to inhibit *P. falciparum* transmission to mosquitos[41]. Here, compound 53, and the chemically related hits 48, 52, and 56, reduced male gamete exflagellation to 20–30% that is considered as an acceptable starting point for not optimized compounds[13]. In SMFA, compound 53 inhibition of parasite transmission after a 24 h treatment at 1 μM was equivalent to that of the positive control MMV0048.

Compound 39, first identified in a HTS on asexual parasites[42] as TCMDC-125769, was subsequently shown active on gametocytes in a parasite ATP assay[7]. Importantly, our work adds that this compound efficiently blocks parasite transmission in mosquito.

Compound 31, a moderate transmission blocker hit, is a singleton. A similar structure was identified in a HTS on 110,000 drug-like compounds as an inhibitor of *P. falciparum* haem detoxification protein HDP involved in hemozoin formation (US20070148185).

The 2,4-diamino-pyrimidine chemotype of compounds 63, 64, and 65 was identified on asexual stages[42] and modifications were introduced to study its proposed kinase inhibitory activity[43]. Our work extended the functional characterization of these compounds to demonstrate activity in blocking parasite transmission.

Importantly, this work also identified, to the best of our knowledge, three novel chemotypes, which ranked amongst the most active compounds tested here in blocking *P. falciparum* transmission through mosquito.

Compound 13 is the second most active compound in SMFA. The chemotype of this singleton has not been previously described in any HTS on malaria parasites. The symmetrical structure in a double 4-piperidinyl benzylic alcohols linked by a short alkyl chain generated an appealing hit with a potential for further optimization. In silico calculation of its physicochemical properties (logD 2.2 and TPSA 47, MW 491.5, 4 HBD, 2 HBA) demonstrated that there is still room for additional exploration without impacting the Lipinski's Rule of Five.

Compound 16, the third most active hit in SMFA with a gametocyte specificity of around 6-fold, is a singleton characterized by an imidazo[1,2-a]pyridine core disubstituted in position 2 and 3. It is unknown if the charged N-oxide pyridine is crucial for the potency or if other cores can be tolerated in the central region. Even if the aromatic thiol might be vital for the geometry of the linker, it is not ruled out that ethers or amines might be allowed. Although this structure has not been identified as an anti-gametocyte hit, interestingly, other imidazopyridines, with a different pattern of substitutions, have been shown to inhibit *P. falciparum* gametocyte viability across several orthologous assays[27].

Compound 69 is another chemotype that was not previously published as an antimalarial hit with a gametocyte specificity of around six-fold. The furanyl-piperidinyl scaffold is not present in any reported transmission blocking drug discovery study. A comparable substituted moiety with distinct decoration on the piperidinyl core is reported in patent number WO2015006752A1 as a p38 MAP Kinase inhibitor VI in combination with other antimalarial drugs. Still, compound 69 looked both small and flexible to be further explored and ultimately optimized.

Prior to our work, a cumulative set of about 300,000 compounds was screened for its activity on *P. falciparum* gametocytes in a variety of assays, making the chemical space explored by gametocyte HTS approximately ten percent of that tested on asexual stages. The present work shows that it is possible to notably extend the amplitude of anti-gametocyte hit identification campaigns and that our screening funnel was further validated by the identification of chemical series that were selected by previous screenings based on orthogonal unrelated assays.

We believe that gametocytes HTS will drive the identification of novel and diverse transmission blocking chemotypes suitable for drug discovery but also chemical probe to interrogate the parasite biology. Furthermore, we not only identified here multiple dual active hits with similar potency against gametocytes and asexual stages, but also molecules preferentially active on gametocytes with the potential to be developed into drugs able to clear residual gametocytes after treatment of malaria episodes or for mass drug administration in asymptomatic populations to hit local transmission hotspots. While future mechanistic studies are warranted for the elucidation of the underlying biology, the availability of novel starting points is extremely relevant for the identification of new drugs to ultimately contribute to malaria elimination and the eradication of parasite.

## Methods

**Compound collection.** CNCCS represents a public-private consortium (www.cnccs.it) which objective is the construction of a collection of compound molecules. In addition to FDA and/or EMA approved drugs, the collection contains a range of chemotypes, from both commercial and non-commercial suppliers, with an optimized structurally diversity (average Tanimoto distance from the nearest neighbour of 0.38; and an average molecular weight of 370 Daltons. The size of the library comprises approximately 120,000 small molecules not biased toward any particular target nor diseases oriented. While the collection was optimized for structural diversity, it maintains an attractive distribution of physicochemical properties (e.g., calculated logD, sp3 character, hydrogen bond donor/acceptors and total polar surface area).

**Compound similarity search.** After hit confirmation, compound similarity searches were performed by generation of circular Morgan fingerprints (radius 2, 2018 bits) for the test compounds using open source RDKit software (http://www.rdkit.org/ release 2014_09_2). The molecular representations generated were used to perform ligand based virtual screening against the target database (i.e., our own screening collection) that is described above or against a subset of the public ZINC database (http://www.zinc.docking.org). Similarity was assessed by the Tanimoto index between the reference and target structures using a cut-off (or threshold) of 0.6. Similar compounds were clustered using Taylor-Butina[32] clustering; a non-hierarchical clustering method that ensures that each cluster contains molecules with a set cut-off distance from the central compound. Compounds selected for purchase or screening follow up were chosen from the most populated clusters, with either the central compound or closed analogues (based on visual inspection) being used to represent the compound cluster. All selected compounds were quality controlled by UPLC-MS prior to testing. The identity and purity of compounds that were not previously associated to antimalarial activity (compound 96, 13, and 16) was verified (Supplementary Fig. 2).

**Parasite lines and culture protocols.** *P. falciparum* asexual parasites of the lines and clones indicated below were cultured in type A/0+ human erythrocytes at 5%

haematocrit and 0.5–10% parasitaemia in RPMI 1640 (Life Technologies). The medium was supplemented with 10% heat inactivated O+ human serum (IBBI, Memphis TN, USA) and 0.36 mM hypoxanthine at 37 °C in modified atmosphere (4% CO$_2$, 3% O$_2$, and 93% N$_2$) by established methods (Trager and Jensen 1976). Gametocyte production was induced by seeding asexual parasite cultures at 0.1% parasitaemia and 5% haematocrit with no further addition of uninfected red blood cells. At the appearance of the oat shaped stage I gametocytes at day 4, 50 mM N-Acetyl Glucosamine (NAG) was added for 3 days to clear residual asexual parasites[44], followed by additional 5 days of culturing in absence of NAG. Parasite lines used in this work were the reference wild type clone 3D7A[45], line 3D7*elo1-pfs16*-CBG99[25], line NF54 *pfs16*-GFP-PyLUC[5], line NF54attB[46] and line NF54 *hsp86*-PpyRE13, produced for this work as described below.

**Production of the *P. falciparum* line NF54 *hsp86*-PpyRE13.** The PpyRE13 luciferase mutant gene from the firefly *Photinus pyralis* was kindly provided by Dr. B. Branchini, Connecticut College, London CT, USA (manuscript in preparation). The multistep cloning strategy to obtain the pCR2.1-attP-*hsp86*-PpyRE13 plasmid, carrying the PpyRE13 red luciferase under the *P. falciparum hsp86* constitutive promoter, was as follows (Supplementary Fig. 1). In brief, the pPpyRE13 plasmid was digested with SpeI and NcoI restriction endonucleases (New England Biolabs) and cloned into a SpeI-NcoI-digested p*hsp86*PenGluc-3'nap plasmid to obtain the intermediate p*hsp86*-PpyRE13 plasmid. Then the p*hsp86*-PpyRE13 plasmid was XhoI-NotI digested to clone the *hsp86*-PpyRE13 cassette into the XhoI-NotI-digested pCR2.1-attP-ULG8-CBG99[47] to finally obtain the pCR2.1-attP-*hsp86*-PpyRE13 plasmid. Parasites from the *P. falciparum* NF54attB line, containing a Bbx1 *attB* site in the *cg6* gene, were grown to the 6% ring parasitemia and transfected with both 100 µg of pCR2.1-attB-*hsp86*-PpyRE13 and 100 µg of pINT[46] plasmids, to obtain the NF54 *hsp86*-PpyRE13 line. Transfection was via electroporation using a BioRad electroporator with 0.31 kV voltage, 960 µF capacitance and resistance to infinity. Double selection started 24 h after transfection by adding 250 µg/ml G418 and 2.5 nM WR99210. After 6 days parasites were treated only with 2.5 nM WR99210 and 3 days later they were allowed to recover in drug free medium. To confirm the integration of the pCR2.1-attP-*hsp86*-PpyRe13 construct into the genome of the NF54-attB strain PCR analyses were performed using Cg6_P1-Plasm_P2 primers and Plasm_P4-Cg6_P3 primers (Supplementary Table 1).

**P. falciparum gametocyte luminescence assay.** Parasite gametocyte luminescence assays were performed in 384-well plates (Thermo, 4332, USA) by infected red blood cells seeded at 40 µL/well at a haematocrit of 0.625% for the strain 3D7*elo1-pfs16*-CBG99, 1.875% for both strains NF54 *hsp86*-PpyRE13 and NF54 *pfs16*-GFP-PyLUC. Assay were performed in the medium: 10.44 g/L RPMI 1640 Medium (Gibco, 51800043), 5.95 g/L HEPES (Sigma, H4034), 50 mg/L Hypoxantine (Sigma, 56700), 10% heat inactivated human serum, 10 ml of Neomycin solution (Sigma, N1142) and 2.1 g/L NaHCO$_3$ (Sigma, 144-55-8). Compounds or DMSO as control were pre dispensed on plates, from 10 mM DMSO stock solutions, via an acoustic droplet ejection device (ATS-100, EDC Biosystems) to achieve the final desired concentration. Assay plates were incubated for 48 h at 37 °C in 4% CO$_2$, 3% O$_2$, and 93% N$_2$ modified atmosphere. After incubation, 20 µL/well of a solution of luciferin (Regis, 1-360242-200) potassium/sodium salt (300 µg/ml) are added to the assay plates and read at suitable luminescence detection after 1 min incubation at room temperature. The whole HTS was performed in 7 rounds of 30–80 plates each in two months' time.

**P. falciparum asexual stage assay.** Parasite proliferation assays were performed in 384-well plates (Thermo, 4332, USA) with a starting parasitemia of 0.25% at a haematocrit of 2% by quantification of parasite lactate dehydrogenase (pLDH). Compounds or DMSO as control were pre dispensed on plates, from 10 mM DMSO stock solutions, via an acoustic droplet ejection device (ATS-100, EDC Biosystems) to achieve de final desired concentration. Assay plates were incubated for 72 h at 37 °C in 4% CO$_2$, 3% O$_2$, and 93% N$_2$ modified atmosphere. After the incubation period, plates are frozen at −80 °C for two h and then thawed at room temperature for 2 h. Then, 50 µL of the developer solution (70 mM Tris-HCl pH 8.0, 0.5% Tween-20, 100 mM Lithium L-Lactate, 125 µM nitro blue tetrazolium, 2 U/ml diaphorase and 100 µM 3-Acetylpyridine Adenine Dinucleotide) are added to each well and let to react for 10 min. Absorbance was then read at 650 nm by a suitable spectrophotometer.

**HeLa cell proliferation assay.** HeLa cells (CCL-2, ATCC, USA) were cultured in DMEM without phenol red (Thermo, 11880, USA) + 10% FBS + 1X PenStrep + 1X Gln. They were plated in a 384-well plate (Thermo, 4334-11, USA) to a density of 2000 cells per well and let recover for 4 h at 37 °C, 5% CO$_2$ in a humidified atmosphere. After the recovery, compounds were transferred to assay plates as *per* compound preparation method. Assay plates were then incubated at 37 °C, 5% CO$_2$ in a humidified atmosphere for 72 h. Cell viability was measured by the CellTiter Glo (Promega, G8080, USA) as *per* manufacturer instruction.

**Male Gamete exflagellation assay.** Asexual parasites of the NF54 line, kindly obtained from Elena Levashina's laboratory (Max Plank Institute for Infection

Biology, Berlin, Germany), were seeded at 1% total parasitemia and grown for 16 days changing medium every day until development of stage V gametocytes (about 1–2% of stage V gametocytemia). 100 µl of gametocyte culture was seeded in each well of a 96 well-plate and 100 µl of 2× concentration of compounds or DMSO as control were added to each well to achieve the desired final concentration. Assay plates were incubated for 48 h at 37 °C in 4% CO$_2$, 3% O$_2$, and 93% N$_2$ modified atmosphere. After the incubation period, 100 µl of medium were removed from each well and the other 100 µl were transferred to a black imaging 96 well plate (ibidi) containing 100 µl of complete medium supplemented with 40 µM of xanthurenic acid. After 15 min, exflagellation events were captured by three ten-second videos each well with an Olympus IX83 inverted microscope with a UPLSAPO 10× N.A. 0.75 objective (Olympus, Japan), equipped with a Hamamatsu ORCA-Flash 4.0 V3 camera. Each video was analyzed using the Icy bio-image analysis software (http://icy.bioimageanalysis.org/) to count the number of exflagellation events.

**P. falciparum standard membrane feeding assay (SMFA).** SMFA was performed commercially by TropiQ Health Sciences (Nijmegen, The Netherlands). Test samples were diluted in DMSO and then in RPMI medium (Gibco, 51800043) to achieve a final DMSO concentration of 0.1%. Diluted samples were combined with stage V gametocytes from *P. falciparum* strain NF54-*hsp70*-luc and incubated for 24 h. Subsequently, human red blood cells and human serum with the sample were added to achieve a haematocrit of 50%. The blood meal was fed to two-day old *Anopheles stephensi* mosquitoes that were starved the night prior to the blood meal. The use of mosquitoes does not require an ethical committee approval. Mosquito infection was analyzed 8 days post-feeding by luminescence measurements of whole mosquitoes as described previously[48]. For each feeder, luminescence intensity was analyzed in 15–24 mosquitoes per feeder ($N = 1$, $n = 15$–$24$). Luminescence background levels were determined from 15–24 uninfected mosquitoes. Controls: vehicle (0.1% DMSO) in duplicate.

**Chemistry.** Unless otherwise stated, all reagents and solvents were obtained from commercial sources and were used as received without further purification. $^1$H spectra were recorded on Bruker AV400 and AV600 spectrometers operating at reported frequencies between 400 and 600 MHz. Chemical shifts for signals corresponding to non-exchangeable protons (and exchangeable protons where visible) were recorded in parts per million (ppm) relative to tetramethylsilane and were measured using the residual solvent peak as reference. UPLC-MS analysis were conducted on a Waters UPLC system with both Diode Array detection, Evaporative Light Scattering Detector and Electrospray (+'ve and –'ve ion) MS detection. The stationary phase was a Waters Acquity UPLC BEH C18 1.7um (2.1 × 50 mm) column. The mobile phase comprised H$_2$O containing 0.1% formic acid (A) and acetonitrile containing 0.1% formic acid (B) with a flow rate 0.5 mL/min.

**Statistics and reproducibility.** All experiments were carried out with at least three technical replicates and multiple compound concentrations. In some Plasmodium reporter assays, due to the amount of compounds tested and the complexity of using human blood, one biological replicate was possible (though against multiple compound concentrations). Compounds scoring positive in these assays were further validated with orthogonal readouts as described in our screening funnel. Data analysis was carried out with Graphpad 9.0.

## Data availability

The data that support the findings of this study are available from the corresponding author upon reasonable request. Source data for figures are provided in Supplementary Data 1.

## Code availability

No custom code or mathematical algorithm were used to analyze the present article data.

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

## Acknowledgements

This work was partially supported by the National Research Council (CNR) - Collezione Nazionale di Composti Chimici e Centro di screening (CNCCS) "Rare, Neglected and Poverty Related Diseases - Malaria Project" and by the grant to P.A. "Italy-South Africa Research Project ISARP2018 "New generation drugs against *Plasmodium falciparum* transmission for malaria eradication" by the Italian Ministries of Health and of Foreign Affairs and International Cooperation. The ISARP2018 project is an Italy's activity in the scope of the Europe & Developing Countries Clinical Trials Partnership – EDCTP2. We gratefully acknowledge Dr. Nardi, Avis Pomezia (Roma) and Prof. Antonio Angeloni, UOC Immunoematologia e Medicina Trasfusionale, Azienda Policlinico Umberto I, Policlinico di Roma, for the gift of human erythrocytes; Dr. Francesco Silvestrini, Dipartimento di Malattie Infettive, Istituto Superiore di Sanità, for advice in the image analysis of the exflagellation assays; Gianluca De Martino, IRBM, for the compounds management.

## Author contributions

A.B., P.A., and G.P. conceived the study, interpreted the results and wrote the manuscript with input from all authors. C.A. collected and analyzed data. G.S. performed and analyzed gametocyte and exflagellation assays and produced the *hsp86*PyRE13 construct. R.G., C.L., O.C., and M.B. performed parasite and HeLa cell assays. A.S. and A.P. did QC on compounds, clustering and analyzed data from a medicinal chemistry point of view. R.L.V. produced the transgenic parasite line *hsp86*PyRE13. M.F. performed microscopy and image acquisition in the exflagellation assay. C.T. reviewed and approved the manuscript. All authors reviewed and approved the manuscript.

## Competing interests

The authors declare no competing interests.

**Additional information**

