## [Peer Review File · Communications Biology]

Reviewers' comments:

Reviewer #1 (Remarks to the Author):

The research article "Novel gametocyte-specific and all-blood-stage transmission-blocking chemotypes from high throughput screening on Plasmodium falciparum gametocytes " reported the establishment of a simple, sensitive and robust HTS assay funnel for the identification/screening of anti-gametocyte compounds including a set of counter-screenings and validation assays. With these tools authors set out to identify bona fide gametocyte active chemotypes and investigated the selectivity of them against the human host and the asexual stage parasites. The whole work is interesting and authors have covered mostly all the stages in transmission blocking. I suggest the addition of few lines in the introduction part about some of the potent groups or moieties found active against gametocyte stage of malaria, supported with the appropriate citations.

Additionally, reviewer will appreciate if author can provide insightful justifications and modifications.

Pg2. Introduction. Line 60, year of the WHO report should be provided.

Pg2. Introduction Line 77 onwards, authors should provide more explanation about TCP profiles with proper citations as they discussed about TPP. New antimalarial candidates can fit different TCPs, including preventing human-to-mosquito transmission (TCP-5), and/or prophylaxis (TCP-4).

Pg2 (Line 82 to 90) Proper citation should be given to the limitation of known drugs.

Pg4 Line 134 please elaborate S/N ratio and its significant values.

List of abbreviations should be added.

Figure 1 is slightly blurred. Authors can provide high resolution images.

Figure legends are not in same formatting especially Figure 5.

Pg13 Line 384, as mentioned here that these compounds 53 and their chemically related hits 48, 52 and 56 reduced male gamete exflagellation to 20 to 30%. Is it a significant inhibition? What about other approved drugs and clinical/ Preclinical molecules?

Pg14 Line 420 correct the number of compounds. Is it 300 or 300000?

Reviewer #2 (Remarks to the Author):

The authors present a novel methodology for high-throughput screening of Plasmodium transmission-blocking compounds, employing bioluminescent gametocytes of the human P. falciparum parasite. They report on the screening of a large number of compounds (~120.000) and on the identification of a few dozen hits either with gametocyte-specific activity or with activity against both sexual and asexual parasite forms. This is a novel approach to the identification of compounds with Plasmodium transmission-blocking potential that deserves being reported to the community. However, I do have a few concerns that the authors should address before publication in Communications Biology can be considered.

Please revise citation formats. For example, the WHO's World Malaria Report 2021 is cited as "2020 1997" (line 60) and a WHO resolution is cited as "Organization 2021" (line 63). Referencing softwares do not always display citations in the most correct manner and care should be exerted to individually inspect each citation for its appropriateness.

Line 67: although widely employed, the term "malaria infections" is incorrect. Malaria is a disease, not an organism. Either state that "Malaria symptoms are..." or that "In Plasmodium infections, ...".

Lines 67-75: when providing these numbers (line 71), the authors should specify whether they are talking about P. falciparum specifically or about all human-infective Plasmodium species.

There are a few grammar / semantic incorrections throughout, which the authors should address.

For example, in Line 78 "malaria acute phase" should be "the acute phase of malaria" and the term "insurgence" (which means "insurrection") is incorrectly used (the authors probably mean "appearance"). In Lines 106 and 126, "To this am" should be "To this end". In Lines 125-126, "paramount important" should be "paramount importance". Line 130, "microscopic examination" should be "light microscopy examination". Line 136, "half 384 well plate" should be "half of a 384-

well plate". Line 145, the authors probably mean "successful" instead of "successive". Lines 208-209, "This procedure gave a collection of 371 clusters whose 205 were singletons" should probably be "This procedure gave a collection of 371 clusters, 205 of which were singletons". And so on...

Paragraph Lines 82-90 makes several statements, none of which is referenced. Please cite appropriate references.

Line 95: it is unclear whether "half of which in only two screenings" refers to the "5 million compounds that were screened against the asexual stage parasites" or the 300,000 compounds that were "tested against *P. falciparum* gametocytes". Likely the latter, but that is exactly the opposite of what the sentence, as it stands, says.

Lines 97-100: the authors write "A variety of HTS gametocyte assays were used so far relying on multiple detection techniques: fluorescent/luminescent reporters, parasite enzymes activity assays, gametocyte ATP content determination, uptake of redox sensitive dyes and the assessment of the motility of male gamete flagella" without providing a single reference. This is totally unacceptable. Each and every one of these reported techniques must be backed by at least one reference.

Lines 100-103: sentence "Beyond assay signal detection procedure incompatible with HTS, the sensitive and specific acquisition of the above assay readouts imposes to purify gametocytes from uninfected erythrocytes, or to cultivate and/or treat them in the absence of human serum" is poorly written and needs to be rephrased. It also needs referencing.

Line 113: the acronym "CNCCS" appears without any definition of what it stands for.

Lines 126-130: the authors state that "... the timing of addition of N-acetyl glucosamine (NAG), used to clear residual asexual stages after appearance of stage I gametocytes, was optimized. A three-day treatment with 50 mM NAG, followed by an additional five days of cultivation, yielded cultures with about 2% gametocytaemia of early-stage V gametocytes, as confirmed by microscopic examination of Giemsa-stained blood smears" and point towards Fig. 1A to substantiate this statement. Fig. 1A does not serve this purpose, unless the authors clearly state that this single image is representative of the entire sample. "hours" should be denoted by "h" throughout and not by either "hours" (e.g. Line 132) or "h" (e.g. Line 137) indistinctly.

Line 134: "S/N ratio" (presumably "signal-to-noise ratio") is not defined or explained.

Line 139: the term "CV%" must be defined and explained. The authors cannot assume all the readers know what this acronym stands for or what this measurement means. Without an explanation for this, Fig. 1C is unintelligible.

Lines 139-140: why are the data of CV% values for the positive and negative controls not shown? Again, without a proper explanation of the meaning of "CV%", the information that both were "less than 10%" is meaningless. The authors should understand that not all the readers will be as versed on this topic as they are, and that explanations of less obvious terms and their meaning must be provided for clarity.

Fig. 1 legend: acronyms must be defined the first time they appear. "RBC" is used without any prior definition.

Lines 158-159: "present at that time" begs the question "when?". This is not clear and the reader is not obliged to know when the assay was performed and what changed in terms of the number of compounds in this library since then.

Lines 189-191: the authors state that "the amount of compounds that confirmed active in both lines (top right field), was similar to the amount of poorly active compounds in both lines (bottom left field)". First, it is not clear from looking at Fig. 3B that this would be the case. Second, why do the authors consider this surprising, i.e., what would they expect to see? Third, can the authors clarify the choice of 40% inhibition for definition of the quadrants in the plot on Fig. 3B?

Line 211: "the activity data on the three parasite strains". These are not three different strains of parasite, but rather 1 transgenic line of strain Pf3D7 and 2 transgenic lines of strain PfNF54.

The authors should mention the work by Azevedo et al. employing bioluminescent *P. berghei* parasites for screening transmission-blocking compounds (PMIDs 28348156, 31752986, 32038528) in their Discussion.

Reviewer #3 (Remarks to the Author):

In this paper the authors screened *Plasmodium* gametocytes against a large compound library. Their premise is that more drugs are needed to prevent the spread of malaria to mosquitos from infected people.

The authors propose the novelty of this paper is 1) the assay using gametocytes and 2) discovery of drugs that block human to mosquito transmission.

From the citations in the methods, it is unclear if this is a new method for screening gametocytes. It also leads me to question if people would be willing to take a medicine to prevent mosquito infection.

From the article, it is not clear if any of these hits will achieve this purpose. Will the 9 compounds that made it through the funnel actually fit this purpose?

The findings are really about the pipeline, not really about the hits themselves. What are the prospects of these hits?

Needs english language editing

Reviewer #1 (Remarks to the Author):

The research article "Novel gametocyte-specific and all-blood-stage transmission-blocking chemotypes from high throughput screening on Plasmodium falciparum gametocytes " reported the establishment of a simple, sensitive and robust HTS assay funnel for the identification/screening of anti-gametocyte compounds including a set of counter-screenings and validation assays. With these tools authors set out to identify bona fide gametocyte active chemotypes and investigated the selectivity of them against the human host and the asexual stage parasites. The whole work is interesting and authors have covered mostly all the stages in transmission blocking.

Comment: The authors would like to thank the reviewer for the appreciation of our work. We are also thankful for the useful comments, and we have done our best to address each of them in the revised text and as outlined below which we feel has strengthen our manuscript.

1. I suggest the addition of few lines in the introduction part about some of the potent groups or moieties found active against gametocyte stage of malaria, supported with the appropriate citations.

Reply: We thank the reviewer for his suggestion that is indeed important to better introduce the subject matter.

We updated the text in order to comply with the reviewer request. Lines 90-93

Additionally, reviewer will appreciate if author can provide insightful justifications and modifications. Pg2.

2. Introduction. Line 60, year of the WHO report should be provided.

Reply: We are sorry for having overlooked this, Year 2021 is now indicated. Line 61

3. Introduction Line 77 onwards, authors should provide more explanation about TCP profiles with proper citations as they discussed about TPP. New antimalarial candidates can fit different TCPs, including preventing human-to-mosquito transmission (TCP-5), and/or prophylaxis (TCP-4).

Reply: We agree with the reviewer's suggestion. Direct references to the profiles relevant to the manuscript (i.e. TCP1, targeting the pathogenic stages, and TCP5, targeting the transmission stages), are now present in the Introduction. Lines 80-83

4. Pg2 (Line 82 to 90) Proper citation should be given to the limitation of known drugs.

Reply: We appreciate the reviewer's input, hence a reference to Adjalley et al. 2011 is now included. Line 85-86

Pg4 Line 134 please elaborate S/N ratio and its significant values.

Reply: We have elaborated the concept and provided more insight on its interpretation. Moreover, thanks to the reviewer's comment we realized that we should have reported the S/B instead of S/N. This is now corrected. Line 136-138

5. List of abbreviations should be added.

Reply: While we understand the reviewer's rationale for a list of abbreviations, the Nature formatting guide does not consider it as part of the manuscript. To comply with the reviewer's comment we inspected the manuscript to make sure that the acronyms are properly defined.

6. Figure 1 is slightly blurred. Authors can provide high resolution images.

Reply: We are sorry for the poor definition of Figure 1. We believe the blurry is due to the editorial management system conversion of our original files. Nonetheless, in the current upload, we introduced high resolution images that should have fixed this issue.

7. Figure legends are not in same formatting especially Figure 5.

Reply: We are sorry for the inconvenience. In the current upload, we introduced high resolution images and associated captions that should now be in the same format.

8. Pg13 Line 384, as mentioned here that these compounds 53 and their chemically related hits 48, 52 and 56 reduced male gamete exflagellation to 20 to 30%. Is it a significant inhibition? What about other approved drugs and clinical/ Preclinical molecules?

Reply: The 20-30% inhibition of male exflagellation must be considered together with the used compound concentration and the fact that these are not optimized molecules (hits). Considering these aspects, a 20-30% inhibition is indicative of active compounds; a threshold of 30% inhibition was set in the HTS based on this assay to select hits (Delves et al Nature Communications, 2018). The discussion section is now expanded to include this comment (Pg14). Lines 312-313

9. Pg14 Line 420 correct the number of compounds. Is it 300 or 300000?

Reply: We are sorry for the typo. The correct number 300,000 is now reported. Line 346

Reviewer #2 (Remarks to the Author):

The authors present a novel methodology for high-throughput screening of Plasmodium transmission-blocking compounds, employing bioluminescent gametocytes of the human *P. falciparum* parasite. They report on the screening of a large number of compounds (~120,000) and on the identification of a few dozen hits either with gametocyte-specific activity or with activity against both sexual and asexual parasite forms. This is a novel approach to the identification of compounds with Plasmodium transmission-blocking potential that deserves being reported to the community. However, I do have a few concerns that the authors should address before publication in *Communications Biology* can be considered.

Reply: The authors would like to thank the reviewer for appreciating the novelty and relevance of our work. Also we would like to thank the reviewer for her/his suggestions that we addressed in the revised manuscript.

1. Please revise citation formats. For example, the WHO's World Malaria Report 2021 is cited as "2020 1997" (line 60) and a WHO resolution is cited as "Organization 2021" (line 63). Referencing softwares do not always display citations in the most correct manner and care should be exerted to individually inspect each citation for its appropriateness.

Reply: We are sorry for the improper references. The reference to the resolution "Organisation 2021" is now eliminated and those referring to the World Malaria Report 2021 and to the Global Technical Strategy for Malaria 2016–2030 are corrected in the introduction and in the references sections.

2. Line 67: although widely employed, the term "malaria infections" is incorrect. Malaria is a disease, not an organism. Either state that "Malaria symptoms are..." or that "In Plasmodium infections, ...".

Reply: We thank the reviewer for pointing out this mistake. "Malaria cases" is now used in the modified sentence (line 61) and the term "infections" was eliminated in the first sentence of the following paragraph (line 68).

3. Lines 67-75: when providing these numbers (line 71), the authors should specify whether they are talking about *P. falciparum* specifically or about all human-infective Plasmodium species.

Reply: We thank the reviewer for pointing out this mistake. Now the sentence reads: "highly differentiated Plasmodium sexual stages that, *in the case of P. falciparum*, mature in red blood cells in 10-12 days". Line 72

4. There are a few grammar / semantic incorrections throughout, which the authors should address. For example, in Line 78 "malaria acute phase" should be "the acute phase of malaria" and the term "insurgence" (which means "insurrection") is incorrectly used (the authors probably mean "appearance"). In Lines 106 and 126, "To this am" should be "To this end". In Lines 125-126, "paramount important" should be "paramount importance". Line 130, "microscopic examination" should be "light microscopy examination". Line 136, "half 384 well plate" should be "half of a 384-well plate". Line 145, the authors probably mean "successful" instead of "successive". Lines 208- 209, "This procedure gave a collection of 371 clusters whose 205 were singletons" should probably be "This procedure gave a collection of 371 clusters, 205 of which were singletons". And so on...

Reply: The authors apologize for the mistakes in the text. We considered all the points raised by the reviewer and corrected them. Further, we inspected the manuscript for similar issues.

5. Paragraph Lines 82-90 makes several statements, none of which is referenced. Please cite appropriate references.

Reply: We apologize for this, relevant references are now present in the paragraph. Lines 84-90

6. Line 95: it is unclear whether "half of which in only two screenings" refers to the "5 million compounds that were screened against the asexual stage parasites" or the 300,000 compounds that were "tested against *P. falciparum* gametocytes". Likely the latter, but that is exactly the opposite of what the sentence, as it stands, says.

Reply: We meant to refer to the "5 million compounds", as indeed the reviewer correctly understood from the original sentence. Lines 96-98

7. Lines 97-100: the authors write “A variety of HTS gametocyte assays were used so far relying on multiple detection techniques: fluorescent/luminescent reporters, parasite enzymes activity assays, gametocyte ATP content determination, uptake of redox sensitive dyes and the assessment of the motility of male gamete flagella” without providing a single reference. This is totally unacceptable. Each and every one of these reported techniques must be backed by at least one reference.

Reply: The author would like to apologize for the missing references. We have updated the manuscript to comply with the reviewer’s comment. Lines 100-103

8. Lines 100-103: sentence “Beyond assay signal detection procedure incompatible with HTS, the sensitive and specific acquisition of the above assay readouts imposes to purify gametocytes from uninfected erythrocytes, or to cultivate and/or treat them in the absence of human serum” is poorly written and needs to be rephrased. It also needs referencing.

Reply: We apologize for the confusion originated by the sentence above. We have rephrased the entire concept in the updated manuscript. Lines 103-106

9. Line 113: the acronym “CNCCS” appears without any definition of what it stands for.

Reply: The reason why there is no definition is that CNCCS is simply the name of the Italian institution holding the collection. The name acronym doesn’t necessarily make a lot of sense in English. Nonetheless the reviewer’s concern is well taken and the acronym is now explained in the text. Lines 116-117

10. Lines 126-130: the authors state that “... the timing of addition of N-acetyl glucosamine (NAG), used to clear residual asexual stages after appearance of stage I gametocytes, was optimized. A three-day treatment with 50 mM NAG, followed by an additional five days of cultivation, yielded cultures with about 2% gametocytaemia of early-stage V gametocytes, as confirmed by microscopic examination of Giemsa-stained blood smears” and point towards Fig. 1A to substantiate this statement. Fig. 1A does not serve this purpose, unless the authors clearly state that this single image is representative of the entire sample.

Reply: The image is indeed representative of the gametocyte cultures used in the HTS. We now make it clear in Fig. 1A caption.

11. “hours” should be denoted by “h” throughout and not by either “hours” (e.g. Line 132) or “h” (e.g. Line 137) indistinctly.

Reply: We are sorry for the missed consistency. The manuscript method section was modified according to the reviewer’s suggestion.

12. Line 134: “S/N ratio” (presumably “signal-to-noise ratio”) is not defined or explained.

Reply: We are sorry for the missed definition. The manuscript was modified to clarify this parameter. Also we realized that the used parameter was S/B (Signal to background). Line 136-138

13. Line 139: the term “CV%” must be defined and explained. The authors cannot assume all the readers know what this acronym stands for or what this measurement means. Without an explanation for this, Fig. 1C is unintelligible.
Lines 139-140: why are the data of CV% values for the positive and negative controls not shown? Again, without a proper explanation of the meaning of “CV%”, the information that both were “less than 10%” is meaningless. The authors should understand that not all the readers will be as versed on this topic as they are, and that explanations of less obvious terms and their meaning must be provided for clarity.

Reply: We apologize for the missing definition of CV% and the missing explanation of acceptable values. The CV% (coefficient of variation %) is a measure of the assay precision and is optimal when less than 10%. The acronym definition and the optimal/acceptable values are new reported. Lines 142-144

14. Fig. 1 legend: acronyms must be defined the first time they appear. “RBC” is used without any prior definition.

Reply: We are sorry for the inconvenience. The acronym is now defined.

15. Lines 158-159: “present at that time” begs the question “when?”. This is not clear and the reader is not obliged to know when the assay was performed and what changed in terms of the number of compounds in this library since then.

Reply: We agree with the reviewer’s concern. We have rephrase to remove the ambiguity. Lines 151-152

16. Lines 189-191: the authors state that “the amount of compounds that confirmed active in both lines (top right field), was similar to the amount of poorly active compounds in both lines (bottom left field”. First, it is not clear from looking at Fig.3 B that this would be the case. Second, why do the authors consider this surprising, i.e., what would they expect to see? Third, can the authors clarify the choice of 40% inhibition for definition of the quadrants in the plot on Fig. 3B?

Reply: We would like to thank the reviewer for raising this point at it gives us the opportunity to avoid the same possible confusion for future readers. To the reviewer’s points:

First, the dots in the upper right quadrant are very tightly dispersed, hence the impression that they are by far fewer than those of the bottom right quadrant. In the updated version we reduced a bit the size of the dots to address this issue. In addition, to avoid any ambiguity, we report the exact number of compounds appearing in the two quadrants in the result section (185 and 331 respectively). Lines 173-174.

Second, we were surprised by the relatively high number of not confirmed hits as all these compounds were active the primary assay (i.e. the screening assay using the 3D7 reporter system) hence we hoped for less false positives using different strains/reporter systems. Nonetheless, we agree with the reviewer that this is probably not a big surprise. We updated the manuscript accordingly.

Third, The 40% threshold was used in line with the 41% threshold calculated to score hit compounds in the primary screening. We revised the text to make this clear

17. Line 211: “the activity data on the three parasite strains”. These are not three different strains of parasite, but rather 1 transgenic line of strain Pf3D7 and 2 transgenic lines of strain PfNF54.

Reply: We agree with the reviewer’s concern. The term “lines” was introduced to replace “strains” where appropriate. Line 194

18. The authors should mention the work by Azevedo et al. employing bioluminescent *P. berghei* parasites for screening transmission-blocking compounds (PMIDs 28348156, 31752986, 32038528) in their Discussion.

Reply: While appreciated the reviewer’s input, we feel that our manuscript is specifically focused on the human parasite, so believe that the references to the assay developed on the rodent malaria model are not as relevant.

Reviewer #3 (Remarks to the Author):

In this paper the authors screened Plasmodium gametocytes against a large compound library. Their premise is that more drugs are needed to prevent the spread of malaria to mosquitos from infected people.

The authors propose the novelty of this paper is 1) the assay using gametocytes and 2) discovery of drugs that block human to mosquito transmission.

1. From the citations in the methods, it is unclear if this is a new method for screening gametocytes. It also leads me to question if people would be willing to take a medicine to prevent mosquito infection.

Reply: While the authors appreciate the reviewer's comment, we believe that we have clarified the scope of the work in the final sentence of the introduction paragraph. Indeed the manuscript proposes an assay funnel (i.e. an ensemble of assays) to identify hits active against gametocytes. By using this approach we demonstrated that it is possible to find molecules with the desired transmission blocking properties.

Regarding the reviewer's observation around the use of potential transmission blocking compounds, we believe that the discussion around this topic is outside the scope of the present work. For this reason we reference the Medicine for Malaria Venture TCP5 (Burrows et al. 2017) in the introduction. In this work the reviewer will find the rationale for the Plasmodium human-to-mosquito transmission blocking approach.

2. From the article, it is not clear if any of these hits will achieve this purpose. Will the 9 compounds that made it through the funnel actually fit this purpose?

Reply: We believe that the results shown in Fig. 5 clearly show that hit compounds fit the purpose. In fact, in this gold standard transmission blocking assay, *P. falciparum* gametocytes treated with each of the nine compounds were unable to successfully transmit the parasite through mosquitoes to the stage of forming the sporozoite-containing oocysts.

3. The findings are really about the pipeline, not really about the hits themselves. What are the prospects of these hits?

Reply: The goal of the work was to demonstrate that the proposed approach is suited for higher efficiency hit identification in the pursuit of malaria eradication through transmission blockage. With this effort, we also discovered new chemotypes fitting the purpose. We believe that the publication of these molecules is important for many in the malaria community to enable mechanistic investigations and drug discovery programs around them.

4. Needs english language editing

Reply: The manuscript was inspected for language improvements.

REVIEWERS' COMMENTS:

Reviewer #1 (Remarks to the Author):

The authors have addressed my comments.

Reviewer #2 (Remarks to the Author):

The authors have satisfactorily addressed my concerns and I consider the manuscripts acceptable for publication in its improved version.